# Development of PPARγ Agonists for the Treatment of Neuroinflammatory and Neurodegenerative Diseases: Leriglitazone as a Promising Candidate

**DOI:** 10.3390/ijms24043201

**Published:** 2023-02-06

**Authors:** Pilar Pizcueta, Cristina Vergara, Marco Emanuele, Anna Vilalta, Laura Rodríguez-Pascau, Marc Martinell

**Affiliations:** 1Minoryx Therapeutics SL, 08302 Barcelona, Spain; 2Minoryx Therapeutics BE, Gosselies, 6041 Charleroi, Belgium

**Keywords:** PPARγ agonist, neurodegenerative disease, leriglitazone, neuroinflammation, mitochondria, clinical trials

## Abstract

Increasing evidence suggests that the peroxisome proliferator-activated receptor γ (PPARγ), a member of the nuclear receptor superfamily, plays an important role in physiological processes in the central nervous system (CNS) and is involved in cellular metabolism and repair. Cellular damage caused by acute brain injury and long-term neurodegenerative disorders is associated with alterations of these metabolic processes leading to mitochondrial dysfunction, oxidative stress, and neuroinflammation. PPARγ agonists have demonstrated the potential to be effective treatments for CNS diseases in preclinical models, but to date, most drugs have failed to show efficacy in clinical trials of neurodegenerative diseases including amyotrophic lateral sclerosis, Parkinson’s disease, and Alzheimer’s disease. The most likely explanation for this lack of efficacy is the insufficient brain exposure of these PPARγ agonists. Leriglitazone is a novel, blood–brain barrier (BBB)-penetrant PPARγ agonist that is being developed to treat CNS diseases. Here, we review the main roles of PPARγ in physiology and pathophysiology in the CNS, describe the mechanism of action of PPARγ agonists, and discuss the evidence supporting the use of leriglitazone to treat CNS diseases.

## 1. Introduction

### 1.1. PPAR Receptors

Peroxisome proliferator-activated receptors (PPAR) are a group of ligand-activated transcription factors belonging to the nuclear receptor superfamily that is expressed ubiquitously throughout the human body. By forming a heterodimer coactivator complex with the retinoid X receptor (RXR), activated PPARs bind to DNA sequences, known as peroxisome proliferator response elements, in target promoter genes to regulate the transactivation of mitochondrial and peroxisome genes involved in multiple protein networks that regulate cellular metabolism and energy homeostasis [1,2,3,4]. In the absence of activating ligands, PPAR–RXR heterodimers are associated with a co-repressor complex that results in the repressed expression of key metabolic genes [4,5]. Three isoforms of PPAR have been identified: PPARα, PPAR β/δ, and PPARγ. All three isoforms play important roles in energy metabolism and storage, but they have different expression patterns and specific functions which have only a limited overlap. PPARα is highly expressed in metabolically active tissues, particularly in the liver, where it plays a major role in fatty acid (FA) metabolism to actively lower lipid levels [5,6]. PPARα also plays a role in ketogenesis by reducing plasma triglyceride levels and increasing high-density lipoprotein levels [4]. PPAR β/δ has a role in FA oxidation in skeletal and cardiac muscle and also helps to regulate blood glucose and cholesterol levels [6]. PPARγ is highly expressed in both white and brown adipose tissue, where it plays a key role in adipogenesis and is a potent modulator of whole-body lipid metabolism and insulin sensitivity [7]; it is also expressed in the skeletal muscle, liver, heart, and intestine [8], and in brain cells such as neurons and glia [9]. The actions of PPARγ include crosstalk with several other important pathways involved in regulating systemic energy homeostasis. For example, PPARγ and the peroxisome proliferator-activated receptor γ coactivator 1α (PGC-1α) play a key role in the co-regulation of mitochondrial oxidative metabolism induction [4].

### 1.2. PPARγ in the Central Nervous System (CNS)

All three PPAR isoforms are also expressed in the brain, where as well as regulating the expression of target genes implicated in glucose, lipid, and energy metabolism, they also promote axonal growth, oligodendrocyte formation and differentiation, and neuronal differentiation [10,11]. Among the three isoforms, PPARγ is the key neuronal isoform in the CNS. It plays a major role in neuroprotection by preventing neuroinflammation, regulates energy homeostasis, and regulates genes involved in FA metabolism including genes for acetyl-CoA carboxylase, FA synthase, and carnitine palmitoyltransferase 1 [12]. Illustrating its role in regulating metabolism, the overexpression of PPARγ in the rodent brain has been shown to increase food intake and abdominal fat [12,13]. 

Under physiological conditions, immunohistochemical studies in rodents and humans have demonstrated the expression of PPARγ in both neurons and glial cells in a number of important brain regions including the prefrontal cortex, basal ganglia, thalamus, and hippocampus [14,15]. Colocalization studies using immunofluorescence have demonstrated that in mice, PPARγ expression is higher in neurons than in astrocytes or microglia across most brain regions; PPARγ colocalization with astrocytes varied between 3.4% in the prefrontal cortex and 26.7% in the nucleus accumbens, and colocalization was consistently low with microglia across the brain regions [15]. Similar results were seen in the human brain with PPARγ showing colocalization with both neurons and astrocytes but not microglia [15]. Interestingly, when a lipopolysaccharide (LPS) was administered to induce a strong neuroimmune response in mice, PPARγ expression was observed in microglia, suggesting that expression may be regulated by the microglial functional state and increased in activated microglial cells [15]. Further evidence to support a role for PPARγ in activated microglia was demonstrated by Song and colleagues who showed that adiponectin regulates the function and polarization of microglia through PPARγ signaling by inducing a switch to the neuroprotective M2 phenotype in response to amyloid β toxicity [16]. These data point towards a potentially important role for PPARγ in mediating anti-inflammatory and neuroprotective effects in the brain [17,18,19].

## 2. PPARγ in Neuroinflammatory and Demyelinating Events

Inflammation is characterized by the activation of macrophages and monocytes at sites of cellular or tissue damage and the subsequent release of proinflammatory cytokines including TNF-α, interleukin (IL-6), and IL-1β, which in turn leads to the stimulation of cyclooxygenase 2 and the production of prostaglandins via the breakdown of arachidonic acid [20]. Indeed, studies have provided evidence that the ligand-mediated activation of PPARγ is associated with a reduction in the accumulation of macrophages and the expression of inflammatory markers in vascular tissue in a mouse model of atherosclerosis [2]. Recently, it has been recognized that PPARγ is a key mediator of the immune response via its ability to inhibit the expression of proinflammatory cytokines, chemokines, and adhesion molecules in peripheral immune cells and resident cells [21], as well as its ability to direct the differentiation of immune cells towards anti-inflammatory phenotypes [5,22].

Neuroinflammation is a protective mechanism that activates pathways necessary to promote the removal of toxic agents released following cellular injury, helping to promote tissue repair and the removal of cellular debris. However, sustained inflammatory responses in the brain are damaging and can inhibit recovery from injury and promote cellular death. Persistent and abnormal neuroinflammatory responses can be triggered by both endogenous and environmental factors. In ischemic stroke, traumatic brain injury, and spinal cord injury, the activation of PPARγ attenuates inflammation by inhibiting the expression of proinflammatory mediators in microglia and macrophages [1]. Following a controlled cortical impact traumatic brain injury in mice, the PPARγ agonist pioglitazone was able to inhibit inflammatory marker genes including TNF-α, inducible nitric oxide synthase (iNOS), IL-1β, and IL-6 and reduce histological damage and inflammation in a dose-dependent manner (reviewed in [17]). Another PPARγ agonist, rosiglitazone, failed to reproduce these effects [23]. In a separate study by Niino and colleagues, the PPARγ agonist troglitazone attenuated the inflammation and clinical symptoms of EAE induced by the administration of myelin oligodendrocyte glycoprotein peptide 35–55 in mice via a postulated reduction in the expression of proinflammatory cytokine genes [24]. In addition, PPARγ is involved in the long-term promotion of cellular and tissue repair and the rescue of brain cells following injury including ischemic stroke, hemorrhagic stroke, traumatic brain injury, and spinal cord injury [1]. Consequently, PPARγ is viewed as an interesting target for therapeutic intervention in patients with brain injuries and neurodegenerative diseases.

Chronic inflammatory damage to the lipid-rich, insulating myelin sheath surrounding axons impairs nerve conduction [25,26]. Demyelination—the pathological process of myelin sheath loss from axons—can be caused by direct injury to the oligodendrocytes [27] that produce the myelin and provide trophic and metabolic support to axons. The oligodendrocyte injury may be amplified by the inflammatory responses to CNS tissue damage, accelerating disease progression. 

Therefore, therapeutic strategies that target oligodendrocytes or enhance the proliferation of OPCs to promote remyelination, and thereby restore signal conduction [28] and functional deficits [29], are attractive in neuroinflammatory and neurodegenerative diseases [30]. PPARγ agonists may protect OPCs by preserving their integrity and by enabling their differentiation into mature myelin-forming cells. Thus, PPARγ agonists have the potential to promote recovery from demyelination through direct effects on oligodendrocytes [20,21,22,23,24,31]. The crucial role of the progression of neuroinflammation in CNS diseases is confirmed with hematopoietic stem cell transplantation (HSCT) procedures. HSCT can halt ongoing inflammatory processes by replacing hematopoietic-originated microglia with donor-derived myeloid precursors. Donor-derived myeloid cells can differentiate into microglia, mimicking their functions and counterbalancing the activated microglia, thereby decreasing neuroinflammation and increasing repair mechanisms such as remyelination [32]. HSCT has been efficacious in halting neuroinflammation in diseases including MS, neuromyelitis optica spectrum disorder, myasthenia gravis, and cerebral adrenoleukodystrophy (cALD) [32]. 

## 3. PPARγ in Brain Metabolism and Bioenergetics

Although the brain accounts for only around 2% of the body weight of an adult human, it is estimated that it uses around 20% of the body’s total oxygen consumption and 25% of the body’s glucose in the resting awake state [33,34]. Most of the energy consumption in the brain, approximately 75–80%, occurs in neurons [35], which require a very large quantity of energy to maintain neuronal resting membrane potentials and to repolarize the cell following electrical activity, including action potentials [36]. Furthermore, neurotransmitter synthesis and release, vesicle recycling, and axoplasmic transport are also highly energy-consuming processes that contribute to energy depletion and the need for a high metabolic rate in neurons [37,38]. The high energy demand of cells in the brain and its dependence on glucose metabolism renders the brain highly vulnerable to impaired energy metabolism, and both hypoglycemia and hyperglycemia can have major effects on cognitive function. Moreover, neurons and neuronal functions are highly sensitive to hypoxia with serious and lasting damage occurring in response to only a few minutes of disruption to the brain’s oxygen supply [39,40]. During the natural aging process, decreases in glucose and oxygen metabolic rates are observed but these can be exacerbated in diseases including Alzheimer’s disease (AD), amyotrophic lateral sclerosis (ALS), Parkinson’s disease (PD), and Huntington’s disease (HD) [41,42]. 

Most of the energy required for these vital neuronal processes is generated locally by the catabolic metabolism of glucose and the mitochondrial production and storage of adenosine triphosphate (ATP). The breakdown of glucose also produces other substrates that are important for biosynthesis. ATP is then used as an energy source for innumerable biochemical reactions within cells that are vital for almost all cellular functions. Cells in the brain, including neurons and astrocytes, can effectively utilize different substrates in addition to glucose as sources of energy including lactate, pyruvate, glutamate, and glutamine [43]. Although both astrocytes and neurons can metabolize similar substrates including pyruvate and glucose [43] and have similar densities of mitochondria [44], under physiological conditions, they use different metabolic pathways [44,45]. For example, FA metabolism plays a solely signaling role in neurons, whereas in glial cells, it is involved both in cellular signaling and in providing energy [46].

Mitochondrial biogenesis is a tightly regulated process involving the growth and division of mitochondria; it is influenced by environmental factors including oxidative stress, cellular division, cellular renewal, and cellular differentiation [47]. It also plays a pivotal role in regulating the cellular response to energy demands and consequently cellular energy availability. The dysfunction of mitochondrial biogenesis is implicated in neurodegenerative disorders and is therefore considered a potentially attractive therapeutic target for these diseases. PPARγ participates in the regulation of mitochondrial function and biogenesis via numerous pathways mainly activating PGC-1α. PGC-1α is the master regulator of mitochondrial biogenesis and plays a central role in coordinating and driving energy metabolism, FA oxidation, gluconeogenesis, glucose transport, glycogenolysis, peroxisomal remodeling, and oxidative phosphorylation. PGC-1α also integrates and coordinates the activity of multiple transcription factors involved in mitochondrial biogenesis including nuclear respiratory factors 1 and 2 (NRF-1 and NRF-2), PPARα, and mitochondrial factor A. Finally, PGC-1α also regulates the expression of several enzymes including superoxide dismutase 1 and 2 (SOD1, SOD2), catalase, and glutathione peroxidase 1 that are involved in antioxidant defense via the detoxification of reactive oxygen species (ROS) [48].

## 4. Thiazolidinediones. Potential for CNS Disorders

### 4.1. Generalities

Thiazolidinediones (TZDs), also called glitazones, are synthetic PPARγ agonists (Table 1); they are potent insulin sensitizers and are the most widely studied PPARγ ligands. The mechanism of action of TZDs was not known until 1995, when Lehmann and colleagues reported that TZDs were high-affinity ligands for PPARγ [49]. They are used as oral hypoglycemic agents to treat patients with type 2 diabetes mellitus (T2DM) and have the potential to limit the risk of developing brain injuries because PPARγ serves as a master regulator of cytoprotective stress responses [1].

In diabetic patients, TZDs have been shown to efficiently reduce the levels of glucose and free FAs in serum, improve excessive lipid accumulation in peripheral tissues, and modulate the expression of adipokines and inflammatory cytokines with an impact on metabolism and whole-body insulin sensitivity [2]. TZDs work by upregulating the c-Cbl-associated protein and the glucose transporter type 4 [50,51], and by regulating tumor necrosis factor α (TNF-α), resistin, and adiponectin, thereby contributing to the oxidation of FAs and TZD-mediated insulin sensitization [52]. Indeed, PPARγ target engagement can be monitored by measuring adiponectin concentration, which is tightly regulated by PPARγ [53,54].

### 4.2. Mechanism of Action

The mechanism of action of TZDs through PPARγ activation has been extensively observed in different articles. The binding of TZD induces a conformational change that alters gene expression of numerous pathways involved in metabolism regulation, including lipoprotein lipase, glucokinase, fatty acyl-CoA synthase, and others [8]. PPARγ agonists improve insulin resistance by increasing adiponectin and GLUT4 expression and opposing the effect of TNF-alpha in adipocytes. Increased GLUT 4 expression will increase glucose uptake in adipocytes and skeletal muscle cells in response to insulin [5]. Although the main target of glitazones is PPARγ, these compounds also inhibit monoamine oxidase (MAO) enzymes at higher doses. Pioglitazone is the most potent MAO-B inhibitor of all glitazones and is selective over MAO-A. These binding properties differentiate pioglitazone from the clinically used MAO inhibitors, which act through covalent inhibition mechanisms and do not exhibit a high degree of MAO-A versus -B selectivity. In MPTP (1-methyl-4-phenyl-1,2,3,6-tetrahydropyridine)-based models, pioglitazone protects monkeys from MPP+ (1 methyl-4-phenylpyridinium)-induced toxicity through the inhibition of MAO-B. MAO-B inhibitors are currently being used as a treatment for PD, but the ones in the market (selegiline, rasagiline) have higher potency (in the nanomolar range) than glitazones [55], which allows to reach at least 80% inhibition required to achieve the desired effects [56].

Moreover, other off-target effects have been reported such as the targeting of the mitochondrial pyruvate carrier (MPC) and acyl-CoA synthetase 4 pathways. PXL-065, the deuterium-stabilized R-enantiomer of pioglitazone, has reduced PPARγ activity compared with other TZDs but retains non-genomic TZD actions including MPC inhibition. PXL-065 has shown positive results in the phase 2 DESTINY trial for non-alcoholic steatohepatitis, producing a significant mean reduction in liver fat content of between 21% and 25% versus the placebo between baseline and 36 weeks [57,58].

### 4.3. Marketed TZDs

Troglitazone was the first TZD approved for the treatment of diabetes, although it was withdrawn later from the market owing to hepatotoxic effects. It was followed by rosiglitazone and pioglitazone (Table 1). Rosiglitazone (Avandia) was first released by GSK in 1999 as a stand-alone drug or in combination with metformin or glimepiride and its use has been controversial for the increased risk of congestive failure. Pioglitazone (Actos) was developed by Takeda, and now it is available as a generic medication for diabetes [59]. These compounds function as potent and selective PPARγ full agonists and are not only highly effective therapies for T2DM but have also aided in further understanding the underlying mechanism by which PPARγ contributes to several physiological processes [53]. These PPARγ agonists have been used in the treatment of hyperlipidemia and hyperglycemia and for non-alcoholic steatohepatitis (NASH) and have been proposed for the treatment of CNS diseases.

Although TZDs are an effective treatment option for patients with T2DM, they have an associated adverse event profile that includes fluid retention, weight gain, bone loss, and congestive heart failure [7,59]. TZDs have been shown to cause dose-related fluid retention causing edema in up to 20% of patients. PPARγ has a critical role in systemic fluid retention through the regulation of renal sodium transport in the collecting duct [60]. In most patients, fluid retention will respond to diuretics such as thiazides. There are reports of an increase in intravascular volume to the point of congestive heart failure, and the risk of cardiac problems is higher with rosiglitazone than pioglitazone [61]. Weight gain is a common adverse effect. TZD agents expand adipose tissue mass via the maturation of preadipocytes into mature adipocytes and increase fat storage by increasing free fatty acid movement into cells. This fat gain occurs primarily in the subcutaneous tissues, sparing the visceral area. Additionally, fluid retention can also increase weight [51].

TZDs can cause bone loss in mice and rats by simultaneously decreasing bone formation (osteoblastogenesis) and increasing bone resorption (osteoclastogenesis), and have been related to a higher rate of fracture in women versus untreated patients [62]. It has been proposed as a possible mechanism that PPARγ could cause the inhibition of osteoblast differentiation and bone formation, promoting osteoclast differentiation [7]. The fracture risk is further increased by additional risk factors, such as postmenopausal females or patients concurrently taking glucocorticoids or proton pump inhibitors (PPIs) [63,64,65].

Pioglitazone has, in some studies, shown correlations with an increased risk of bladder cancer. This effect varies in a duration-dependent and dose-dependent fashion. Additionally, most recent analyses do not support an increased risk. Rosiglitazone was not associated with an increased risk of bladder cancer in any analysis, suggesting the risk is drug-specific and not a class effect [66].

In contrast, TZDs potentially have anti-cancer properties. Some studies demonstrated that the activation of PPARγ receptors could induce cancer cell apoptosis [67]. This controversial effect could be because of PPARγ may be involved in both the tumor-suppressive and oncogenic roles of PPARγ in bladder cancer [68,69,70]. Recently, it has been suggested that PPARγ may be a favorable prognostic factor in patients with bladder cancer. The study proposes that the transactivation of PPARγ could be served as a potential strategy for the chemoprevention and therapeutic treatment of bladder cancer [71].

### 4.4. Neuroprotective Effects of TZDs

Neuroprotective mechanisms involving PPARγ TZD in neurodegenerative diseases have been extensively studied in preclinical models. PPARγ agonists have demonstrated efficacy in a range of animal models for CNS inflammatory and neurodegenerative disorders, including AD [72], PD [73,74], HD [75], ALS, and Friedreich’s ataxia (FRDA) [76]. For example, in a mouse model of AD, treatment for 7 days with pioglitazone resulted in a reduction in the number of activated microglia and reactive astrocytes in the hippocampus and cortex compared with control mice [77]. Pioglitazone has also demonstrated the ability to reduce neurodegeneration and neuroinflammation in two models of experimental autoimmune encephalomyelitis (EAE), a highly reproducible and well-established preclinical model used in multiple sclerosis (MS) research [78], and to improve motor function in an ATP binding cassette subfamily D member 1/2 (*Abcd1*/*Abcd2)* double knockout mouse model of ALD [79]. Pioglitazone was able to improve mitochondrial dysfunction and oxidative stress in an induced model of Huntington’s disease [80]. Rosiglitazone induced neuronal mitochondrial biogenesis and improved glucose utilization in mice in an apolipoprotein E-independent manner leading to improved cellular function and providing further evidence for PPARγ as a therapeutic target in AD [81]. Moreover, in a mouse model of T2DM, rosiglitazone increased the expression of brain-derived neurotrophic factor (BDNF) and induced synaptic plasticity by enhancing long-term potentiation at the hippocampal synapses; the transient transfection of a constitutively active form of PPARγ induced the increased expression of BDNF and ionotropic AMPA and NMDA glutamate receptors and promoted dendritic spine formation [82]. In models of Parkinson’s and Huntington’s diseases, rosiglitazone has been found to reduce mitochondrial dysfunction and promote antioxidant defense [83], and to restore mitochondrial function and reduction in ROS [82,84,85]. In ALS models, troglitazone has demonstrated the ability to improve the survival of motor neurons in rats, whereas delayed disease onset and protected against motor neuron degeneration in a mouse carrying an ALS-related mutation [86,87].

Additionally, the activation of PPARγ using TZDs has also been demonstrated to impact mitochondrial biogenesis. They can induce mitochondrial biogenesis via the activation of PGC-1α in human subcutaneous adipose tissue [88]; enhance the ability of cells to maintain their mitochondrial potential [89]; induce mitochondrial biogenesis and glucose utilization leading to improved cellular function [81]; and increase astroglial and neuronal glucose uptake helping to restore brain ATP levels and inhibit oxidative damage after stress [90,91].

PPARγ agonists have been indicated as neuroprotective agents, supporting synaptic plasticity and neurite outgrowth. For these reasons, many PPARγ ligands have been proposed for the improvement of cognitive performance in different pathological conditions such as autism, schizophrenia, Parkinson’s disease, and Alzheimer’s disease [83,92]. For instance, a paper described how rosiglitazone improved neurocognitive deficits depending on aging in older animals [93]. They showed that, at the behavioral level, acute and chronic rosiglitazone administration increased learning ability. In parallel, synaptic plasticity in the dentate gyrus of rosiglitazone-treated rats was restored. It is, in fact, previously reported that rosiglitazone improves cognitive function by enhancing dendritic spine density in specific brain regions [94], probably increasing mitochondrial biogenesis and function, thus improving synaptogenesis and memory formation [94]. This evidence would support the use of TZDs as mediators of neuroprotection, ameliorating neuroinflammation, mitochondrial function, and neurotrophin levels, thus constituting a putative treatment for cognitive decline associated with neurological diseases.

The use of either PPARγ antagonists, such as GW9662 [95,96], or partial agonists [97] or conditional neuron-specific PPARγ knockout mice [98] reversed the protective effect of the fully PPARγ agonist pioglitazone and other TZDs confirming that this protective effect is due to the PPARγ activity.

In primary cultures of rat oligodendrocyte progenitor cells (OPCs), PPARγ agonists have also demonstrated the ability to promote the differentiation of progenitor cells and enhance their antioxidant defenses [31]. The results obtained in astrocytes treated with the PPARγ agonist pioglitazone suggest an induced alteration of astrocyte metabolism by increasing the glucose flux through GLUT-1 protein, increasing lactate production, and inducing mitochondrial membrane hyperpolarization [90].

With the exception of ischemic stroke, for which several large trials have reported promising outcomes with pioglitazone and rosiglitazone [1,99], the results from clinical trials for PPARγ agonists in CNS disease have not typically lived up to the promising preclinical data. In AD, a positive effect of treatment with rosiglitazone was observed in a small cohort of patients [100], but this was not replicated in several larger clinical trials of the drug [101,102]; similar results were obtained in placebo-controlled trials of pioglitazone. Additionally, preliminary studies with pioglitazone in small cohorts of young autism patients showed that the treatment was well-tolerated and shows a potential signal in measures of social withdrawal, repetitive, and externalizing behaviors [103]. Pioglitazone was able to control behavioral symptoms of autism, mainly due to the different effects of activated PPARγ, such as a reduction in the brain inflammatory response and an increase in mitochondrial function [104,105]. Pioglitazone has also been tested in clinical trials for ALS, PD, and FRDA but did not achieve significant efficacy for the treatment of any of the diseases, including in a trial that combined pioglitazone with riluzole in patients with ALS [106]. The failure of TZD agonists in clinical trials of CNS disorders in humans may be due to inadequate target exposure in the CNS owing to the low penetrability of these drugs across the blood–brain barrier (BBB). Although TZDs can affect PPARγ in the CNS, TZDs seem to insufficiently cross the BBB [1]. In contrast to the preclinical studies where the tested doses were very high, the administered doses in clinical studies were the usual doses indicated for the treatment of T2DM. These doses are inadequate to attain a therapeutic concentration in the brain, which would point to the primary reason for the limited success of pioglitazone in these clinical trials [107]. This conclusion is supported by the fact that the only neurodegenerative disease for which pioglitazone has shown effective anti-inflammatory effects to this date is in patients with metabolic syndrome and in relapsing remitting MS, a condition in which the BBB is known to be disrupted [108]. Although these results might suggest the beneficial effects of PPARγ agonism in other cerebral inflammatory conditions, information on the potential efficacy of pioglitazone in conditions in which the BBB is intact is lacking. It is likely that significantly higher doses of pioglitazone, perhaps greater than five times the highest current labeled dose, would be required in such conditions [109]. As the administration of higher doses cannot be recommended due to the adverse effects, novel compounds should be developed to ensure that the target concentration will reach the CNS [107].

Leriglitazone hydrochloride (5-[[4-[2-[5-(1-hydroxyethyl)pyridin-2-yl] ethoxy] phenyl] methyl]-1,3-thiazolidine-2,4-dione hydrochloride) is a novel, neuroprotective, brain-penetrant PPARγ full agonist consisting of the hydrochloride salt of the active metabolite M4 of pioglitazone (Figure 1; Table 1). In contrast to pioglitazone and other TZDs, leriglitazone shows the ability to cross the BBB to reach the CNS at effective and safe concentrations. So far, it is the first TZD compound that has been proven to reach the brain in a sufficient concentration to engage that target effectively [109]. When the PPARγ agonist activity of leriglitazone was tested in a transactivation assay, the half maximal effective concentration (EC50) was 9 µM, and there was no evidence of PPARα or PPARδ agonist activity [109]. This EC50 value could not represent the true potency of the drug, as the transactivation assay that mimics what happens in vivo in the nucleus is greatly dependent on the method used (e.g., co-activator, reporter genes, specific cells), and there are big differences among the reported values of known PPAR gamma agonists. As a consequence, in vitro PPAR gamma agonism transactivation assays are only valid to characterize the relative potency among different compounds rather than providing a correct estimation of their potency relative to other cellular targets [110]. Leriglitazone indeed shows efficacy in relevant CNS cells in concentrations between 10–500 nM [109].

Moreover, leriglitazone better stabilized (LBD) regions including the AF-2 coregulator surface and a region of helix 7 that includes a ligand-dependent SUMOylation site (K367) implicated in promoting the PPARγ-mediated repression of pro-inflammatory genes, affording also slightly better transcriptional efficacy than pioglitazone [111].

In preclinical studies, leriglitazone has shown a good PK profile with very high bioavailability in mice, rats, and dogs (85–90%), and a 50% increase in the brain-to-plasma exposure ratio compared with pioglitazone in mice.

Leriglitazone has been validated in in vivo and in vitro preclinical models that cover most of the potential known effects of PPARγ receptor activation, with a particular focus on CNS indications. Recent studies using leriglitazone have also demonstrated a potential role for PPARγ in promoting axonal myelination, providing neuroprotection in both neurons and astrocytes, and preserving the integrity of the BBB [109,112].

## 5. Using Leriglitazone for the Treatment of Neurodegenerative Diseases

Given the neuroprotective potential of PPARγ treatment and the ability of leriglitazone to cross the BBB, leriglitazone has been investigated as a novel therapeutic option for patients with neurodegenerative diseases. Leriglitazone has demonstrated efficacy in several preclinical models of CNS diseases such as ALD and FRDA [109,113]. ALD is a chromosome X-linked, rare, inherited, neurodegenerative disease resulting from the loss of function of the encoded ALD protein (ALDP), an ABCD transporter located in the peroxisomal membrane. A deficiency of ALDP impairs peroxisomal β-oxidation of very-long-chain FAs (VLCFAs), leading to their accumulation in plasma and tissues, particularly the brain, spinal cord, and adrenal glands [114,115,116]. Additionally, mutations in the *ABCD1* gene cause changes in adhesion molecules and tight junctions in the brain endothelium, which in turn promote an increase in BBB permeability independently of VLCFA accumulation [112]. FRDA is a rare autosomal recessive neurodegenerative disease characterized by progressive spinocerebellar and sensory ataxia, cardiomyopathy, diabetes mellitus, and skeletal deformities [117]. FRDA is caused by a decrease in the expression of the mitochondrial protein frataxin to 5–30% of the normal levels, most commonly as a result of large expansions of GAA triplet repeats in the first intron of the *FXN* gene [118], although point mutations leading to FRDA have also been described in rare cases. Frataxin is a ubiquitous, highly conserved mitochondrial protein that is involved in iron homeostasis and metabolism and iron–sulfur cluster (ISC) biogenesis.

A phase 2 trial of leriglitazone in men and women with FRDA (FRAMES; NCT03917225) [119] and a phase 2/3 trial of leriglitazone in men with AMN (ADVANCE; NCT03231878) have recently been completed [120]. A phase 2/3 open-label clinical study (NEXUS) in pediatric patients with early-stage cerebral ALD (cALD) (EUDRACT number 2019-000654-59) is currently in progress in Europe. In the following section, we describe the specific mechanisms by which leriglitazone might provide therapeutic benefits for patients with neurodegenerative diseases and summarize the available evidence for the use of leriglitazone (Figure 1).

### 5.1. Halting Neuroinflammation

Only a limited additional inflammatory component is present in AMN, which is characterized by slowly progressive adult-onset spinal cord axonopathy with associated demyelination leading to spastic paraparesis [121]. As well as the AMN phenotype, approximately 60% of male patients with ALD will develop additional progressive cerebral demyelination and cerebral neuroinflammation over their lifetime, known as cerebral ALD (cALD) [122]; onset can occur either in childhood or during adulthood. This phenotype of progressive cALD exhibits rapid and severe cerebral demyelination frequently with the disruption of the BBB and the subsequent infiltration of immune cells, mainly monocytes/macrophages and CD8^+^ T cells, into the brain. Brain inflammatory demyelination results in severe cognitive and neurological disability leading to a vegetative state within 2–5 years from the onset of clinical symptoms, and death [123]. Currently, there are no approved pharmacological treatments for ALD. HSCT is performed in pediatric patients with early-stage cALD, although it is associated with serious and sometimes fatal complications and is not generally performed in adult patients who develop cALD. Recently an alternative for cALD treatment, eli-cel, which uses ex vivo transduction with the Lenti-D lentiviral vector (LVV) to add functional copies of the ABCD1 gene, has been approved by the FDA (June 2022).

PPARγ agonism activated by leriglitazone can also regulate inflammatory pathways by the transrepression of the proinflammatory transcriptional factor nuclear factor κ-light-chain-enhancer of activated B cells (NF-κB) [109]. In a model of ALD using rat spinal cord motor cell cultures, NF-κB pathway activation was assessed using Western blot immunoassays of nuclear factors of the κ light polypeptide gene enhancer in B cell inhibitor α (IKBα, an inhibitor of NF-κB). In this assay, leriglitazone reduced NF-κB pathway activation after injury with VLCFA as indicated by increased IKBα protein. Furthermore, in the supernatants of the same cultures, the concentration of IL-1β, a downstream target of NF-κB, was markedly reduced, further demonstrating that leriglitazone can counter inflammatory responses. These preclinical findings are supported by data from a phase 1 clinical trial that showed evidence of CNS target engagement (increase in adiponectin levels) and changes in inflammatory biomarkers in plasma and cerebrospinal fluid (CSF). Accordingly, leriglitazone decreased the concentrations of several proinflammatory cytokines and chemokines (IL-8, IL-6, interferon-γ-inducible protein 10, and monocyte chemoattractant protein-1) in the CSF and plasma of healthy volunteers, likely by repressing NF-κB activation [109].

In an *Abcd1*/*Abcd2* double knockout mouse model of ALD, 6 months of treatment with leriglitazone, given daily at three increasing doses, dose-dependently improved motor function, including improvements from baseline in the balance beam and rotarod test at all doses, and reduced inflammation in spinal cord tissues [109]. Furthermore, immunohistochemical analyses of mouse spinal cord demonstrated that at the highest dose of leriglitazone, amyloid precursor protein (a marker of axonal swelling) and MAC-3 antigen levels fell, indicating reduced axonal degradation and decreased microglial activation, respectively [109]. Neuroinflammation is an important hallmark of cALD, the most severe form of ALD that involves cerebral demyelination with a rapid loss of neurological function after the onset of initial symptoms [124]. Leriglitazone also reduced the neurological disability observed in an EAE mouse model of neuroinflammation in a dose-dependent manner [109].

Altogether these results support leriglitazone as a potential effective treatment for both phenotypes of ALD: AMN, which is associated with progressive non-inflammatory axonopathy resulting in myelopathy, and cALD, characterized by progressive inflammatory cerebral demyelinating lesions. Moreover, leriglitazone treatment could be extended to other diseases with neuroinflammatory components.

### 5.2. Leriglitazone in Demyelination

Genetic defects affecting glia are the main cause of primary demyelination in leukodystrophies, whereas inflammatory damage to myelin and oligodendrocytes is the main cause of neuroinflammatory diseases such as MS and cALD [125,126].

In ALD, saturated VLCFAs accumulate in the blood and CNS and trigger the demyelinating inflammatory response [127]. Although demyelinating brain lesions develop only in patients with cALD, myelin-related abnormalities in the spinal cord and brain are also present in AMN patients. It is plausible that the development of the severe cerebral phenotype may be associated with an immune response that targets oligodendrocytes and abnormal myelin with an excess of VLCFAs, thus resulting in demyelination, reactive gliosis, impaired oligodendrocyte differentiation, and aberrant immune activation in affected patients [128].

In vitro evidence indicates that leriglitazone protects oligodendrocytes after VLCFA injury and increases the phagocytosis of myelin debris by microglia [109], a process that is necessary to initiate remyelination. Moreover, in vivo evidence from the cuprizone demyelination model shows that leriglitazone increases myelination. Electron microscopy revealed that mice receiving leriglitazone had proportionately more myelinated axons after 3 weeks of treatment than untreated controls [109]. Of note, myelination increased in this mouse model at a dose which provides an exposure of leriglitazone that is similar to that observed at the doses assessed in ALD clinical trials [109]. Together, these results suggest that leriglitazone has direct actions on oligodendrocytes that could prevent demyelination or even promote remyelination and reduce proinflammatory status in ALD. To explore whether the promising in vitro and in vivo results translate to remyelination in humans, future clinical studies could assess changes in the expression of specific markers for OPCs and mature oligodendrocytes over time in leriglitazone-treated patients with ALD. Diffusion tensor imaging (DTI) has recently been proposed as a sensitive measure for evaluating disease progression outcomes in patients with AMN [129]. DWI/DTI techniques are more advanced variants of MRI that also include MRS and perfusion imaging [130]. These techniques measure the integrity of tissue using two types of measures: fractional anisotropy (FA) and mean diffusivity (MD) or apparent diffusion coefficient (ADC). Decreases in FA and increases in MD or ADC are considered markers of neuronal fiber loss and reduced gray and white matter integrity. Brain DTI enables the quantitative regional assessment of cerebral white matter microstructure in routine clinical practice [131,132]. Radial diffusivity on DTI is related to myelin content and could be appropriate for assessing neuroprotection in patients with ALD once the first signs of demyelination have been detected on a brain MRI.

In addition, neurofilament light chain (NfL) has been recently used and validated as a sensitive biomarker indicative of axonal damage in several CNS disorders. Axonal damage associated with demyelination and inflammatory processes leads to the release of neurofilaments into the extracellular space, which reach the CSF and the peripheral blood. Hence, NfL-increased levels in plasma have been related to axonal damage and neuronal death. Notably, plasma levels of NfL markedly increase when patients convert to cALD, and these levels strongly associate with Loes scores, with patients with more advanced cALD presenting both higher NfL levels and higher Loes scores [133]. Thus, elevations of plasma NfL can be associated with progression in patients with cALD. Along the same lines, a recent publication in pediatric patients with cALD found an increase in plasma NfL levels compared with controls. A significant correlation was shown between the plasma and CSF levels of NfL, and a very good association was also found between the plasma NfL level and the MRI Loes score. Moreover, 9 of the 11 patients with cALD who were assessed pre- and post-HSCT showed a decrease in plasma NfL levels at 1 year post-HSCT [134]. In the same line, the results from the ADVANCE clinical study on plasma biomarker data showed that NfL levels were significantly increased at week 96 in placebo patients with cerebral lesion progression, supportive of a drug effect on reducing axonal degeneration by leriglitazone [120].

### 5.3. Protecting BBB Integrity

The BBB ensures the optimal homeostasis of the brain’s internal environment. Its anatomical structure, consisting of the gliovascular complex with tight junctions between the endothelial cells, enables the highly selective transport of substances from the blood to the brain [135,136]. Although the BBB serves a critical protective role, it does pose a challenge regarding cerebral drug delivery for the treatment of neurodegenerative diseases.

BBB dysfunction is a critical element of brain aging and neurodegenerative diseases, where BBB integrity is compromised as a consequence of the breakdown of endothelial cells and cell junctions [137]. The upregulation of adhesion molecules and other proinflammatory molecules in brain endothelium is necessary to allow activated monocytes to adhere and transmigrate, and as a result, leak through the BBB. The resulting entrance of macrophages to the brain aggravates neuroinflammation and demyelination. Such BBB disruption with the migration of leukocytes into the brain has for a long time been implicated as crucial to disease progression in patients with cALD [138]. In vitro investigations of the ability of leriglitazone to decrease the adhesion of monocytes to brain endothelial cells have been conducted to understand the potential impact of leriglitazone on BBB disruption. In these studies, leriglitazone demonstrated an ability to decrease the adhesion of monocytes to endothelial cells [109]. Furthermore, in monocyte-derived macrophages from patients with ALD, leriglitazone treatment in vitro resulted in the cellular population being less skewed towards an inflammatory phenotype, together with a decrease in the proinflammatory cytokine TNF-α levels to 80–90% in the THP-1 monocyte cell line following inflammatory-induced exposure to LPS [109]. Altogether these results suggest a mechanism that could allow leriglitazone to prevent the pathological disruption of the BBB and demonstrate its anti-inflammatory potential [109].

In ALD, the gold-standard scoring system to evaluate the radiological progression of disease was proposed by Loes in 1994 [138]. The Loes score provides a determination of the extent of brain damage by rating the severity of white matter lesions on MRI. Scores range from 0 (normal) to 34 (abnormal). Loes score is routinely utilized in disease follow-up and provides guidance on therapeutic decisions regarding HSCT. In clinical practice, Loes score, gadolinium enhancement, and lesion volume are used as surrogates for rapid clinical evolution and overall survival in this disease, and to assess progression to cALD. There is abundant literature showing that MRI outcomes are highly predictive of the further progression of MRI lesions and clinical deterioration [139,140,141,142,143,144,145].

Increased matrix metallopeptidase 9 (MMP-9) in serum, plasma, and CSF is observed in several neurodegenerative diseases such as MS and AD, and in neuroinflammatory conditions such as traumatic brain injury and stroke. MMP-9 has been associated with the disruption of BBB integrity in ALD patients [112], and elevated MMP-9 in plasma has been reported in boys with cALD [146]. The progression to cALD is characterized by the inability to appropriately resolve the inflammatory reaction to ensuing insults by the brain’s immune system. Monocyte skewing towards the proinflammatory state and increased BBB permeability precede overt demyelination in ALD [112,147]. An additional biomarker implicated in cALD is macrophage inflammatory protein-1β (MIP-1β; also known as chemokine [C-C motif] ligands 4), a proinflammatory chemokine shown to be elevated in the CSF of boys with cALD [148]. The results from the ADVANCE clinical trial on patients with ALD revealed that leriglitazone significantly reduced the progression of cerebral lesions and myelopathy symptoms such as balance deterioration while preserving the BBB integrity, as observed through a significant reduction in the plasma levels of MMP-9 [120].

### 5.4. Amelioration of Mitochondrial Dysfunction and Improved Cellular Glucose Metabolism

Mitochondrial dysfunction can be associated with mitochondrial respiratory chain complex deficiency and the capacity of neurons to generate ATP. The resulting energy failure state is coupled with an increase in energy demand by the demyelinated axon and is therefore particularly relevant in tracts such as corticospinal tracts that have long projection axons. Therefore, pharmacological pathways whose modulation may improve mitochondrial function, increase antioxidant defense, and reduce ROS generation can protect against axonal degeneration and may be important therapeutic targets [48].

The improvements in mitochondrial bioenergetics produced by PPARγ agonists have shown the capacity to induce neuroprotective and restorative effects in neuronal cells in preclinical models of neurodegenerative diseases [149] including ALS [87], PD [150], FRDA [76], AD [151], and AMN [152], which is the spinal cord-related manifestation of ALD. Mitochondrial dysfunction, oxidative stress, and bioenergetics failure play major roles in the pathogenesis of ALD.

The ability of leriglitazone to reverse mitochondrial alterations associated with ALD has been established in in vitro and in vivo models. In motor dysfunction models of ALD (*Abcd1* knockout and *Abcd1/Abcd2* double knockout mice that recapitulate the AMN phenotype), the administration of leriglitazone restored bioenergetic function and ATP concentrations, increased the expression of PGC-1α and NRF-1 via the activation of the PPARγ–PGC-1α pathway, and exerted neuroprotective effects in neurons and astrocytes following VLCFA-induced toxicity [109]. In addition, leriglitazone augmented the transcription of the important mitochondrial function and oxidative stress-related genes *NRF1* and *SOD2* and demonstrated evidence of target engagement with FA-binding protein 4 and PPARγ in patients with ALD [109]. Furthermore, leriglitazone restored markers of oxidative stress and improved mitochondrial function in ALD in in vitro models of spinal cord motor neurons treated with VLCFA [109].

Patients with FRDA have abnormal glucose homeostasis, disrupted pyruvate metabolism, and defects in the β-oxidation of FAs that contribute to mitochondrial metabolic dysfunction and a reduction in bioenergetic capacity. FRDA is driven by a significant decrease in the expression of frataxin and it has been demonstrated that the PPARγ–PGC-1α pathway is dysregulated because of this deficiency [76]. Leriglitazone has multiple effects on dorsal root ganglia frataxin-deficient neurons: increasing cell survival and frataxin protein levels and reducing neurite degeneration and α-fodrin cleavage mediated by calpain and caspase 3. Additionally, leriglitazone restored the mitochondrial membrane potential and partially reversed the decreased levels of the mitochondrial Na^+^/Ca^2+^ exchanger, resulting in improved mitochondrial function and better calcium homeostasis. In frataxin-deficient cardiomyocytes, leriglitazone prevented lipid droplet accumulation and improved FA β-oxidation and energy metabolism without increases in frataxin levels. This could be linked to a lack of significant mitochondrial biogenesis and cardiac hypertrophy. Furthermore, leriglitazone improved motor function deficit in a commonly used model for FRDA, YG8sR mice. Leriglitazone significantly increased the markers of mitochondrial biogenesis in cells derived from patients with FRDA. These results would suggest that targeting the PPARγ pathway with leriglitazone may be effective in treating FRDA by increasing mitochondrial function and biogenesis and increasing frataxin levels in compromised frataxin-deficient dorsal root ganglia neurons. This supports the use of leriglitazone as a potential treatment for FRDA [113].

In clinical practice, MRS is a non-invasive neurochemical technique that measures biological metabolites in target tissues that have been used in studies of brain aging, neurodegeneration, and cognition. Among several metabolites that can provide information related to metabolism and bioenergetics, two main metabolites that often show alterations in patients with neurodegenerative diseases are N-acetylaspartate, a marker of neuronal integrity, and mIns (MyoInositol), which reflects the extent of glial cell proliferation and neuronal damage. Neurodegeneration can also be successfully measured by cerebral perfusion [130]. In the FRAMES clinical trial, a numerical difference between the placebo and the leriglitazone-treated group was seen in the tNAA/mIns concentration ratio assessed by magnetic resonance spectroscopy (MRS) [119].

In addition, these findings show that leriglitazone, through the activation of PPARγ receptors, increases the number of mitochondria in neurons, reduces oxidative stress, and improves the regulation of FA β-oxidation to recover the energetic function of these cells. The effects of leriglitazone may be expanded in other neurodegenerative diseases involving mitochondria deficiency.

### 5.5. Decrease in Iron Accumulation

Cerebral iron accumulation occurs in several neurodegenerative diseases including PD, FRDA, MS, and the heterogenous group of conditions collectively known as neurodegeneration with brain iron accumulation (NBIA) disorders. Iron accumulation is the result of the downregulation of the three major pathways of mitochondrial iron utilization: ISC biogenesis, heme synthesis, and mitochondrial iron storage. ISC proteins assist vital biological processes such as enzymatic catalysis, DNA synthesis and repair, ribosome biogenesis, iron homeostasis, and heme synthesis [153]. Thus, mitochondrial ISC synthesis is key to the maintenance of numerous vital enzymatic activities as well as the maintenance of cellular iron homeostasis. Because mitochondrial ISCs are vital for cellular iron homeostasis, the disruption of ISC biogenesis and function may result in a compensatory increase in cellular iron uptake and the mitochondrial targeting of iron to compensate for deficiencies in ISC and heme synthesis [153]. The marked increase in iron uptake and targeting of the mitochondria leads to the iron loading of this organelle. Hence, the dysregulation of ISC biogenesis plays a role in the pathophysiology of neurodegenerative diseases [153].

In FRDA, a loss of frataxin results in decreased ISC biogenesis and the dysregulation of iron levels. The consequent downregulation results in mitochondrial iron accumulation and dysfunction, decreased ATP production, free radical accumulation, increased oxidative stress, and downstream cell death [154]. The iron accumulation, mitochondrial dysfunction, and reactive species overproduction caused by frataxin deficiency can additionally affect glial cells in patients with FRDA, leading them to assume phenotypes that exacerbate neuron loss [154]. The mechanisms of frataxin loss and ISC biogenesis have been widely studied. Rötig and colleagues reported a deficient activity of the ISC-containing subunits of several mitochondrial respiratory complexes and mitochondrial aconitase, an enzyme of the tricarboxylic acid cycle, in patients with FRDA [155]. Numerous subsequent studies have supported the involvement of frataxin in ISC biogenesis and have demonstrated interactions between frataxin and the ISC assembly complex [156,157]. Frataxin has also been described as acting as an iron chaperone protein that is required for the reversible modulation of aconitase activity [158] and may function as an iron donor in vitro [159]. In biopsies from patients with FRDA, frataxin deficiency has been associated with iron deposits, mitochondrial dysfunction, and ROS production [155,160,161,162,163]. Progressive dentate nucleus abnormalities are evident in vivo in FRDA, and the rates of change of iron concentration and atrophy in these structures are sensitive to the disease stage. Novel MRI techniques, such as quantitative susceptibility mapping (QSM) or the T2* transverse relaxation time, provide a measure of heavy metal loading in the brain by measuring magnetic susceptibility [164]. Using QSM in a 2-year study, Ward and colleagues found an increase in iron concentration in patients with FRDA [165]. In addition, MRI QSM showed a progressive increase in dentate nucleus iron concentration in a longitudinal study in patients with FRDA [165]. Iron accumulation might also be related to the dysfunction of neural cells caused by alterations in mitochondrial activity, lipid metabolism, membrane remodeling, and autophagy [166]. These findings are consistent with an increased iron concentration and atrophy early in the disease, followed by iron accumulation and stable volume in later stages. This pattern suggests that iron dysregulation persists after the loss of the vulnerable neurons in the dentate [165].

PPARγ agonists can act to induce ISC biogenesis and iron homeostasis through PGC-1α induction. Thus, PPARγ agonists are good candidate drugs to treat iron accumulation and improve mitochondrial function in patients with neurodegenerative diseases. The processes of iron accumulation triggered by ISC biogenesis dysfunction outlined above make FRDA a suitable starting point for investigating the effect of PPARγ agonists on iron accumulation. However, attempts to study iron accumulation in cellular and mouse models of FRDA have failed because of the inability of these models to mimic the increase in iron deposits seen in patients with FRDA [165].

A more promising avenue to investigate the effects of PPARγ agonists on iron accumulation has been in the field of NBIA disorders. These are a group of inherited neurological disorders in which iron accumulates predominantly in the structures of the basal ganglia, including the globus pallidus and substantia nigra, resulting in brain MRI changes. The cortex and cerebellum can also be affected in the most severe NBIA disorder subtypes [166,167,168]. Several studies have shown defective iron metabolism in different NBIA disorder models, revealing an association between pantothenate kinase 2 (PANK2) deficiency, causing one of the most common subtypes of NBIA disorders called pantothenate kinase-associated neurodegeneration (PKAN) and the impairment of heme synthesis and ISC biogenesis, the two mitochondrial iron-dependent pathways [169,170]. More recently, researchers have been able to replicate the human PKAN phenotype by demonstrating that PKAN neurons were less viable in the presence of induced pluripotent stem cell (iPSC)-derived astrocytes with accumulated iron [171]. Such models provide a suitable tool for studying the effect of leriglitazone on iron metabolism. To evaluate the protective effect of leriglitazone against iron accumulation, we have used iPSC-derived astrocytes from patients with NBIA disorders treated with leriglitazone, showing an improvement in the morphology of and significantly decreasing iron accumulation in iPSC-derived NBIA astrocytes carrying different disease mutations, without affecting cell viability [172].

These results have implications for other neurodegenerative disorders that might be targeted with PPARγ agonists. Astrocytes constitute the scaffold of the entire CNS, and their processes participate in the neurovascular unit of the BBB. Their functions range from regulating cerebral blood flow to maintaining water, small-molecule, and neurotransmitter homeostasis, forming synapses, and supporting neuron metabolism. Astrocytes may participate in iron accumulation and aberrant redistribution in other disorders, including FRDA [154].

The FRAMES phase 2 study further expands on the possible role of leriglitazone in treating iron accumulation in patients with FRDA. In this study, the assessment of biochemical MRI QSM indicated no further iron accumulation in the cerebellum (dentate nuclei) of patients [119].

## 6. Clinical Development of Leriglitazone

### 6.1. Phase 1 Study: Tolerability and PK Profile

A phase 1, randomized, double-blind, placebo-controlled, single-center study in healthy male volunteers was conducted to determine the safety, tolerability, PK profile, brain concentration, and food effect of leriglitazone and was completed in November 2016. The results indicated that leriglitazone was rapidly absorbed after oral administration, and overall exposure was not affected under fed conditions. The PK profile showed low variability and a dose-dependent increase in drug exposure. Detectable concentrations of leriglitazone and adiponectin in human CSF confirmed the ability of leriglitazone to penetrate the brain and engage the PPARγ target in the CNS independently of the integrity status of the BBB. PPARγ target engagement, as indicated by plasma adiponectin levels, increased in a dose-dependent manner with increases in the plasma adiponectin level of 200% at 135 mg and 450% at 270 mg [173]. These increases in adiponectin levels are markedly higher than what has been previously attained with the highest approved dose (45 mg/day) of pioglitazone, which induced an 80% increase in plasma adiponectin level after 4 months of treatment. These data indicate that, whereas it is not possible to achieve sufficient target engagement in the brain within the recommended dose range of pioglitazone, leriglitazone can achieve sufficient target engagement [109]. In addition, leriglitazone decreased the proinflammatory cytokine concentration in human plasma and CSF [109], possibly through the repression of NF-κB activation [109]. These promising results provided support for the clinical development of leriglitazone in the phase 2/3 ADVANCE clinical trial.

### 6.2. FRAMES Study: Efficacy, Safety, and PK in Pediatric and Adult Patients with FRDA

Overall, 39 patients were enrolled (mean age 24 years; 43.6% women; mean time since symptom onset 10.5 years): 26 patients received leriglitazone (20 completed), and 13 received a placebo (12 completed). There was no difference between the groups in the spinal cord area from baseline to week 48. The iron accumulation in the dentate nucleus assessed by MRI QSM (quantitative susceptibility mapping) was greater on the placebo group, and a numeric difference was seen in the tNAA/mIns concentration ratio assessed by magnetic resonance spectroscopy (MRS). These measures have been proven to be very sensitive and reflect neurochemical abnormalities in upper limb ataxia such as in spinocerebellar ataxia type 1 (SCA1) [174]. The most frequent adverse event was peripheral edema [119]. The improvement in the treated group is consistent with the QSM results, suggesting that leriglitazone at least partially corrects metabolic deficits, as shown in preclinical FRDA models [119]. The limitations of the study include the fact that a proportion of patients had advanced disease at baseline and therefore had possibly already reached a plateau of spinal cord atrophy. The sample size was small, and the observation time was short; this limited somewhat the power of the analyses and the conclusions that can be drawn regarding clinician-reported and patient-reported outcomes.

In conclusion, although the primary endpoint was not met in this study, results from secondary endpoints provide evidence for clinical proof of concept for the use of leriglitazone in patients with FRDA and support assessment in larger studies.

### 6.3. ADVANCE Study: Efficacy, Safety, and PK in Adult Patients with ALD

A phase 2/3, randomized, double-blind, placebo-controlled study in men with AMN and no active cerebral lesions was conducted in the USA and Europe (ADVANCE). ADVANCE was discussed with regulators as a pivotal study for drug registration. A total of 116 patients were randomized at a ratio of 2:1 (77 to leriglitazone and 39 to placebo). The objective was to evaluate the effect of leriglitazone on myelopathy and on cerebral progression, as well as safety, after 96 weeks of treatment.

Leriglitazone was generally well tolerated, and the results from this clinical trial have been published in the Lancet Neurology journal [120]. The study did not meet the primary outcome for the 6 min walk test. However, the clinical measures of body sway (assessing balance) demonstrated clinically relevant differences, and favorable trends were observed for the Expanded Disability Status Scale (EDSS), the Severity Score System for Progressive Myelopathy (SSPROM), and quality of life. In addition, the results also showed that leriglitazone reduces the progression of cerebral lesions and only placebo group patients developed clinically progressive cALD (6 out of 39 patients).

### 6.4. NEXUS Study: Efficacy, Safety, and PK in Pediatric Patients with cALD

NEXUS is an open-label study in boys aged 2–12 years which will evaluate whether leriglitazone can arrest cerebral progression in pediatric patients with cALD. NEXUS began in February 2020 and is ongoing. NEXUS will follow patients for 96 weeks, with an interim assessment at 24 weeks. NEXUS is recruiting patients with ALD and the first evidence of cerebral MRI lesions. All patients will receive leriglitazone and they will be monitored for safety and for changes in clinical symptoms and cerebral MRI lesions.

## 7. Future Research

The available data support the use of brain-penetrant PPARγ agonists such as leriglitazone in ALD and FRDA [109,113,119], and more generally in other neuroinflammatory and neurodegenerative conditions. Studies in other preclinical models such as AD, PD, HD, stroke, NBIA disorders, cognitive disorders associated with neurodegenerative diseases, mitochondrial diseases, or MS support leriglitazone as a potential therapy for these neurodegenerative diseases. The next aims should be focused on investigating new indications using specific preclinical models and new imaging and biochemical biomarkers in neurological disorders before and after leriglitazone treatment.

Although the etiology of other CNS diseases may differ from ALD and FRDA, they share common altered pathways and pathophysiology features such as neuronal loss, axonal damage, oxidative stress, and mitochondrial dysfunction, which can be aggravated by neuroinflammation in a vicious cycle involving microglia activation and the disruption of the BBB. Thus, leriglitazone therapy treatment might be extended to a broader range of neurodegenerative diseases with a high unmet medical need such as MS, ALS, PD, and AD.

## 8. Conclusions

The evidence covered by this review indicates that PPARγ agonists act simultaneously on several cellular metabolism and repair processes in the CNS that become dysfunctional because of cellular damage caused by neuroinflammatory and neurodegenerative diseases. Leriglitazone shows superior BBB penetration and a favorable safety profile, allowing interaction with PPARγ in the CNS above the level that can be safely achieved with pioglitazone and other TZDs and thereby offers potential benefits in treating CNS diseases. The ability of leriglitazone to readily penetrate the BBB holds the promise that the beneficial effects observed in preclinical models of CNS diseases will translate to neuroprotective effects that could halt or reverse disease progression in people with devastating neurodegenerative and neuroinflammatory conditions. This is particularly important in diseases such as ALD, where, besides the motor dysfunction phenotype, patients can also develop the devastating cerebral form, or in other neuroinflammatory diseases associated with BBB disruption. The ability of leriglitazone to access the CNS even when the BBB remains intact gives the potential to halt inflammatory and demyelinating processes in patients with neurodegenerative diseases. To establish the clinical benefit of leriglitazone in other neurodegenerative and neuroinflammatory diseases the corresponding clinical trials should be performed.

## Figures and Tables

**Figure 1 ijms-24-03201-f001:**
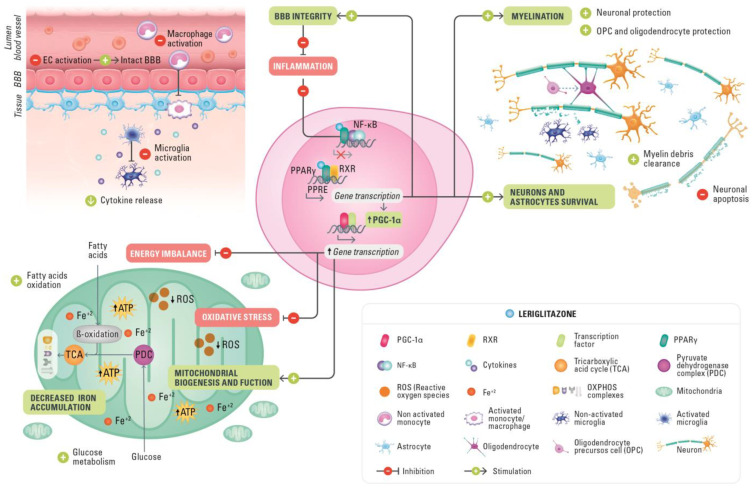
Mechanism of action of leriglitazone in neuroinflammatory and neurodegenerative diseases. ATP, adenosine triphosphate; BBB, blood–brain barrier; EC, endothelial cells; Fe^+2^, iron; NF-κB, nuclear factor κ-light-chain-enhancer of activated B cells; OPC, oligodendrocyte precursor cell; OXPHOS, oxidative phosphorylation system; PDC, pyruvate dehydrogenase complex; PGC-1α, PPARγ coactivator-1α; PPARγ, peroxisome proliferator-activated receptor γ; PPRE, peroxisome proliferators response elements; ROS, reactive oxygen species; RXR, retinoid X receptor; TCA, tricarboxylic acid cycle.

**Table 1 ijms-24-03201-t001:** PPARγ agonist thiazolidinedione drugs marketed or actively in development.

Drug Name	Commercial Name	Developed by	Structure	Mechanism of Action	Indication
Rosiglitazone	Avandia	GlaxoSmithKline	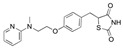	PPARγ agonist	Type 2 diabetes mellitus
Pioglitazone	Actos	Takeda	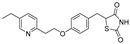	PPARγ agonist	Type 2 diabetes mellitus
Leriglitazone	Nezglyal	Minoryx Therapeutics	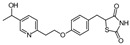	PPARγ agonist	Being developed for X-linked adrenoleukodystrophy and FRDA

FRDA, Friedreich’s ataxia; PPAR, peroxisome proliferator-activated receptor.

## Data Availability

Not applicable.

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
