# Peer review of "Development of PPARγ Agonists for the Treatment of Neuroinflammatory and Neurodegenerative Diseases: Leriglitazone as a Promising Candidate"

_ijms, 2023, doi:10.3390/ijms24043201_

Round 1
Reviewer 1 Report
In the manuscript titled “Development of PPARγ agonists for the treatment of neuroinflammatory and 1 neurodegenerative diseases; leriglitazone as a promising candidate”, Pizcueta and colleagues have provided an overview of the roles played by PPARs, esp. PPARγ, in the physiology and bioenergetics of the CNS as well in pathological states, introduced TZDs as PPARγ agonists and summarized their beneficial and adverse effects. They have then focused on leriglitazone, a TZD manufactured by their company Minoryx, as a promising drug for managing CNS disorders.
Major concerns:
- The premise of the review is that the failure of thiazolodinediones in clinical trials is most probably due to “low penetrability of these drugs across the BBB” in humans. But the authors fail to justify why the same medications were efficacious in animal models. They state “pioglitazone has been shown to cross the BBB of mice… only at low concentrations”, but the fact that it still does exert considerable effects in mice negates their claim of a dosage issue and BBB penetrability. In the same vein, the fact that rodents have basal metabolic rates seven times greater than that of humans, shows in spite a much lower bioavailability, and thereby less amounts crossing the BBB in rodents, TZDs are still efficacious as opposed to human clinical trials. Authors need to explicitly explain this discrepancy.
- Section “4. Thiazolidinediones. Potential for CNS disorders” should break down into subsections. There is no logical flow to the sentences/paragraphs, just citing articles from here and there. The section lacks coherence in the present form and it is not clear what message the authors attempt to convey.
-Authors have failed to cite some relevant research and review papers, examples include:
WIN55212-2 attenuates amyloid-beta-induced neuroinflammation in rats through activation of cannabinoid receptors and PPAR-γ pathway. PMID: 22634229
Involvement of PPAR receptors in the anticonvulsant effects of a cannabinoid agonist, WIN 55,212-2. PMID: 25448777
PPARs in the central nervous system: roles in neurodegeneration and neuroinflammation. PMID: 28220655
Minor comments:
- Authors tend to simply state findings of others articles rather than actually interpreting them, which is the main aim of a review article. Examples include but are not limited to lines 79-81. The cited finding should be scientifically discussed.
- HSCT has been used a couple of times in the text before being spelled out in line 422.
- Language correction, the list is exhaustive but some examples include:
Line 132; arresting should change to “stop” or “halt”
Line 75; a lipopolysaccharide (LPS) injection was administered should change to “lipopolysaccharide (LPS) was administered”
Line 139; is accounted for neurons should change to “occurs in neurons”
Line 225-226; the sentence should be rephrased as it is grammatically wrong.
Line 241; omit “drugs” and “the” coming before further
Line 252; “in” should change to “with”
Line 264; “shown” should change to “has shown”
- Punctuation correction:
Line 238; close the parenthesis after Actos
Line 270; PPAR-gamma should be written “PPARγ”
Line 358-354; the sentence is way too long, should break up into a few shorter ones.
- Define PGC-1α
Author Response
Point to point response
1 reviewer
Major concerns:
- The premise of the review is that the failure of thiazolodinediones in clinical trials is most probably due to “low penetrability of these drugs across the BBB” in humans. But the authors fail to justify why the same medications were efficacious in animal models. They state “pioglitazone has been shown to cross the BBB of mice… only at low concentrations”, but the fact that it still does exert considerable effects in mice negates their claim of a dosage issue and BBB penetrability. In the same vein, the fact that rodents have basal metabolic rates seven times greater than that of humans, shows in spite a much lower bioavailability, and thereby less amounts crossing the BBB in rodents, TZDs are still efficacious as opposed to human clinical trials. Authors need to explicitly explain this discrepancy.
Thank you for the comment.
The discrepancy between the failure of pioglitazone in CNS clinical trials , it is mostly due to the higher doses used in the preclinical models with pioglitazone that cannot be translated to the clinical trials since the approved doses for diabetes are up to 45 mgr clearly lower that the ones effectives in CNS. Although it is true that the metabolism of the pioglitazone is higher in rodents, the experimental procedures in preclinical models compensate this issue either with several administrations during the day or with compound mixed in the feed to guarantee continuous drug levels or giving higher doses. A part of the different pharmacokinetics between humans and rodents, the percentage of unbound fraction of the drugs in both plasma and brain which are the effective available drug also changes depending on the species. For instance in humans to achieve the same effect than in mouse in plasma it could be increased 3 times. From Rodriguez-Pascau, SciTranslational Medicine 2021.
Based on the preclinical data and having in mind that the permeability to the CNS is around 8% with pioglitazone (see below) the effective dose for CNS disease should be much higher than the dose in diabetes. Unfortunately, it could not be tested due to the limitations of the adverse events but in the cases such as when the blood brain barrier is disrupted in for instance multiple sclerosis and pioglitazone can be reach the CNS easily, the compound is effective in improving the BBB integrity (Negroto et al 2016).
For the sake of clarity we also change this point in the manuscript
The failure of TZD agonists in clinical trials of CNS disorders in humans may be due to inadequate target exposure in the CNS owing to the low penetrability of these drugs across the BBB. Although TZDs can affect PPARγ in the CNS, TZDs seem to insufficiently cross the BBB [48].. In contrast to the preclinical studies where the tested doses were very high, the administered doses in clinical studies were the usual doses indicated for the treatment of T2DM. These doses are inadequate to attain a therapeutic concentration in the brain, which would point to the primary reason for the limited success of pioglitazone in these clinical trials [109].
Since the administration of higher doses cannot be recommended due to the adverse effects, novel compounds should be developed to ensure that the target concentration will reach the CNS [109].
- Section “4. Thiazolidinediones. Potential for CNS disorders” should break down into subsections. There is no logical flow to the sentences/paragraphs, just citing articles from here and there. The section lacks coherence in the present form and it is not clear what message the authors attempt to convey.
Thank you for the comment.
We have restructured this section and divided in the following subsections
Generalities
Mechanism of action
Marketed TZDs
Neuroprotective effects of TZDs
-Authors have failed to cite some relevant research and review papers, examples include:
WIN55212-2 attenuates amyloid-beta-induced neuroinflammation in rats through activation of cannabinoid receptors and PPAR-γ pathway. PMID: 22634229
Involvement of PPAR receptors in the anticonvulsant effects of a cannabinoid agonist, WIN 55,212-2. PMID: 25448777
PPARs in the central nervous system: roles in neurodegeneration and neuroinflammation. PMID: 28220655
The papers are included references 17,18 and 19
Minor comments:
- Authors tend to simply state findings of others articles rather than actually interpreting them, which is the main aim of a review article. Examples include but are not limited to lines 79-81. The cited finding should be scientifically discussed.
Thank you for the advice. We first tried to show the findings published in the papers, compiling all information available and at the end of the paragraphs to conclude and discuss scientifically. We have tried to be objective and to avoid overinterpretation.
- HSCT has been used a couple of times in the text before being spelled out in line 422.
corrected
- Language correction, the list is exhaustive but some examples include:
Line 132; arresting should change to “stop” or “halt”
done
Line 75; a lipopolysaccharide (LPS) injection was administered should change to “lipopolysaccharide (LPS) was administered”
done
Line 139; is accounted for neurons should change to “occurs in neurons”
done
Line 225-226; the sentence should be rephrased as it is grammatically wrong.
done
Line 241; omit “drugs” and “the” coming before further
done
Line 252; “in” should change to “with”
done
Line 264; “shown” should change to “has shown”
done
- Punctuation correction:
Line 238; close the parenthesis after Actos
done
Line 270; PPAR-gamma should be written “PPARγ”
done
Line 358-354; the sentence is way too long, should break up into a few shorter ones.
corrected
- Define PGC-1α
peroxisome proliferator-activated receptor γ coactivator 1α, define in Page 2
Reviewer 2 Report
In their interesting review, the authors describe the role played by PPARγ in central nervous system physiological processes, particularly in cellular metabolism and repair. They comprehensively describe the mechanisms of action of PPARγ agonists and the evidence supporting the use of leriglitazone for the treatment of CNS diseases. The ability of leriglitazone (unlike pioglitazone and other TZDs) to penetrate the BBB would play a neuroprotective effect capable of counteracting the progression of neurodegenerative and neuroinflammatory diseases.
The review is well done and skillfully organized. The bibliography is exhaustive and covers the scientific literature on the matter.
I believe the review is worthy of publication.
Author Response
Thank you for your nice comments
Reviewer 3 Report
The manuscript is a well-written review of the current status of PPARγ agonists in the treatment of several chronic neuroinflammatory and neurodegenerative conditions, specifically reviewing the range of actions and possible ameliorative effects of leriglitzone.
The information is well-presented. Their bias and conflict of interest is clearly disclosed. I do have some concerns with the heavy emphasis on Ref. 111 as authoritative, however, the limitations of that study are noted and the authors' intent is to indicate that as a potential springboard for future research, not as a conclusive study. Even given that, the quality of the review content is high, and there is a useful discussion of several potential avenues of future research. Consequently, I would recommend publications with some minor revisions, noted below.
Line 337. '…. TZDs seem to cross poorly the BBB.” is awkward. Suggest “ …. TZDs seem to inefficiently cross the BBB.”
Line 351. Please note word spacing issues.
Line 355. suggest changing 'the target' to 'that target'.
Line 365-366. Confusing sentence which needs to be rewritten for clarity.
Line 426. add a 'the' in front of FDA.
Line 458. Delete 'the' before 'clinical practice'.
Lines 464-465. Suggest a change to ' In the FRAMES clinical trial, a numerical difference was observed between …..'.
Lines 674-676. remove bold underlining.
* Fig 1 is elegant, but much of the print is impossible to read, even when magnified. This should be reworked to be readable.
Author Response
Point to point review
Reviewer 3
The manuscript is a well-written review of the current status of PPARγ agonists in the treatment of several chronic neuroinflammatory and neurodegenerative conditions, specifically reviewing the range of actions and possible ameliorative effects of leriglitzone.
The information is well-presented. Their bias and conflict of interest is clearly disclosed. I do have some concerns with the heavy emphasis on Ref. 111 as authoritative, however, the limitations of that study are noted and the authors' intent is to indicate that as a potential springboard for future research, not as a conclusive study. Even given that, the quality of the review content is high, and there is a useful discussion of several potential avenues of future research. Consequently, I would recommend publications with some minor revisions, noted below.
Thank you for your comments
Line 337. '…. TZDs seem to cross poorly the BBB.” is awkward. Suggest “ …. TZDs seem to inefficiently cross the BBB.”
done
Line 351. Please note word spacing issues.
corrected
Line 355. suggest changing 'the target' to 'that target'.
done
Line 365-366. Confusing sentence which needs to be rewritten for clarity.
We rewrote the paragraph and we added some information to clarify it
Line 426. add a 'the' in front of FDA.
done
Line 458. Delete 'the' before 'clinical practice'.
done
Lines 464-465. Suggest a change to ' In the FRAMES clinical trial, a numerical difference was observed between …..'.
Thanks, changed
Lines 674-676. remove bold underlining.
Done
* Fig 1 is elegant, but much of the print is impossible to read, even when magnified. This should be reworked to be readable.
The Figure has been changed to one with higher resolution
Round 2
Reviewer 1 Report
No further comments